# Forensic Analysis of Novel SARS2r-CoV Identified in Game Animal Datasets in China Shows Evolutionary Relationship to Pangolin GX CoV Clade and Apparent Genetic Experimentation

**Adrian Jones [1]** , **Steven E. Massey [2],*** , **Daoyu Zhang [3]** , **Yuri Deigin [4]** and **Steven C. Quay [5]**

1   Independent Bioinformatics Researcher, Melbourne, VIC 3000, Australia
2   Biology Department, University of Puerto Rico-Rio Piedras, San Juan, PR 00925, USA
3   Independent Genetics Researcher, Sydney, NSW 2120, Australia
4   Youthereum Genetics Inc., Toronto, ON L4J 8G9, Canada
5   Atossa Therapeutics, Inc., Seattle, WA 98104, USA
*   Correspondence: steven.massey@upr.edu

**Abstract:** Pangolins are the only animals other than bats proposed to have been infected with SARS-CoV-2 related coronaviruses (SARS2r-CoVs) prior to the COVID-19 pandemic. Here, we examine the novel SARS2r-CoV we previously identified in game animal metatranscriptomic datasets sequenced by the Nanjing Agricultural University in 2022, and find that sections of the partial genome phylogenetically group with Guangxi pangolin CoVs (GX PCoVs), while the full RdRp sequence groups with bat-SL-CoVZC45. While the novel SARS2r-CoV is found in 6 pangolin datasets, it is also found in 10 additional NGS datasets from 5 separate mammalian species and is likely related to contamination by a laboratory researched virus. Absence of bat mitochondrial sequences from the datasets, the fragmentary nature of the virus sequence and the presence of a partial sequence of a cloning vector attached to a SARS2r-CoV read suggests that it has been cloned. We find that NGS datasets containing the novel SARS2r-CoV are contaminated with significant *Homo sapiens* genetic material, and numerous viruses not associated with the host animals sampled. We further identify the dominant human haplogroup of the contaminating *H. sapiens* genetic material to be F1c1a1, which is of East Asian provenance. The association of this novel SARS2r-CoV with both bat CoV and the GX PCoV clades is an important step towards identifying the origin of the GX PCoVs.

**Keywords:** pangolin; coronavirus; SARS-CoV-2; game animal; forensic; metagenomics; Guangxi

## 1. Introduction

A zoonotic jump from animals has been proposed as a potential origin for SARS-CoV-2 [1]. The virus emerged in Wuhan in late September/early October [2] to as late as mid-October to mid-November 2019 [3] and spread worldwide, leading to over 6 million deaths to date [4]. The Huanan Seafood Market (HSM) was implicated early in the pandemic as a potential source of the virus, ostensibly via zoonosis [5], but several of the earliest reported cases had no link to the market [6], and importantly no animals at the HSM were found to test positive for the virus [7]. Furthermore, several early COVID-19 market cases may have occurred via human to human transmission rather than via a zoonotic jump [8]. In addition, the presence of human infections associated with lineage B at the market [9] makes a market origin less likely, as lineage B is likely a derived lineage, while lineage A is likely ancestral [10] (lineage A and lineage B are the first two major SARS-CoV-2 lineages to emerge in 2019, and are only separated by two single nucleotide variants (SNVs) [11]). An alternative hypothesis that the progenitor of SARS-CoV-2 was present in one of the laboratories in Wuhan conducting bat coronavirus research and accidentally escaped has not been sufficiently investigated [12].

Before the COVID-19 outbreak, bat-SL-CoVZC45 and bat-SL-CoVZXC21 were the only two published coronaviruses related to SARS-CoV-2 (hereafter SARS2r CoVs) [13,14]. These

sequences were obtained from *Rhinolophus pusillus* bats in Zhoushan city between June 2015 and February 2017, and were isolated through the direct inoculation of homogenized bat intestinal material into the brains of 3-day old suckling Balb/c mice [13]. However, the bat-SL-CoVZC45 (also termed ZC45) and bat-SL-CoVZXC21 (also termed ZXC21) genomes are only 89.12% and 88.65% identical to SARS-CoV-2 respectively. Shortly after the publication of the SARS-CoV-2 genome, the RaTG13 genome was published [15], with a significantly higher nucleotide identity to SARS-CoV-2 (96.14%). The virus was sampled from a mineshaft in Mojiang, Yunnan where in 2012 six miners may have been infected with a SARS related coronavirus, killing three [16].

In addition to RaTG13, the 3 other closest relatives to SARS-CoV-2 were sampled from Southern Yunnan [5,17] and Northern Laos [18], which are located 1500 km and 1700 km from Wuhan respectively. Definitive routes for SARS-CoV-2 travel to Wuhan have not been identified via either natural zoonosis or research related sampling, although a research conduit existed for the transportation of SARS2r CoVs from Southern China to Wuhan. Extensive sampling of bats and bat fecal material in Yunnan has been conducted by researchers affiliated with the Wuhan Institute of Virology (WIV), Wuhan University and Guangdong Institute of Applied Biological Resources (GIABR) [19,20] (Figure 1).

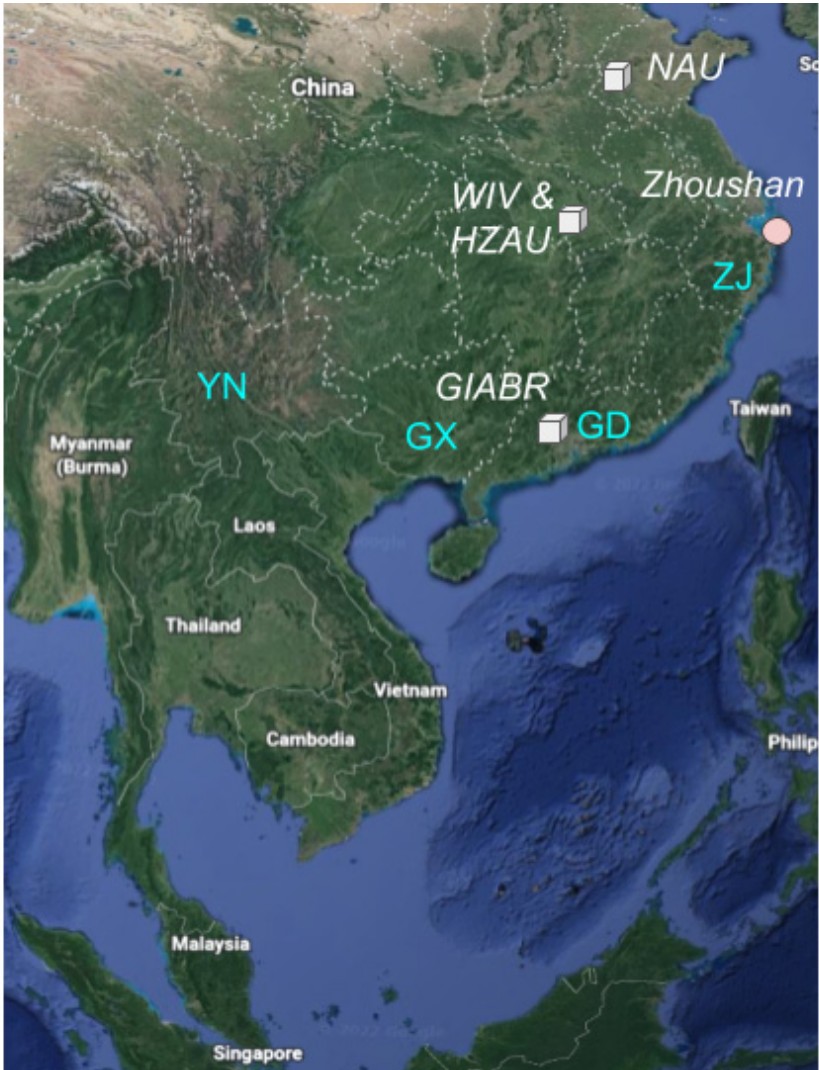

**Figure 1.** Location of sequencing centers, provinces and sampling locations discussed here. Sequencing centers are represented as cubes with text in italics: Nanjing Agriculture University (NAU); Wuhan Institute of Virology (WIV), Huazhong Agricultural University (HZAU); Guangdong Institute of Applied Biological Resources (GIABR). Provinces in blue text: Guangdong (GD); Guangxi (GX);

Yunnan (YN); Zhejiang (ZJ); sampling locations are represented as pink circles: Zhoushan city, Zhoushan Island, Zhejiang province, China.

In addition to bat hosted SARS2r-CoVs, two pangolin hosted SARS2r-CoV clades have been proposed: Guangdong (GD) and Guangxi (GX) pangolin coronaviruses (PCoVs). GD PCoV was first identified on the 31 January 2020 by Wong [21] as SARS-CoV-2-related, by finding that the receptor binding domain (RBD) amino acid sequence of a novel coronavirus found in pangolin organ tissues sequenced by Liu et al. [22] at the GIABR had high homology to the RBD of SARS-CoV-2. Consequently, it has been proposed that SARS-CoV-2 acquired its RBD via recombination with, or from an ancestor in common with a GD PCoV [23]. However, the SARS2r-CoV in the Liu et al. datasets may be contamination related rather than pangolin hosted given the low number of SARS2r-CoV reads, presence of human genomic sequences, presence of non-pangolin hosted virus sequences in similar abundance as SARS2r-CoV sequences, and correlation of of SARS2r-CoV sequences with high bacterial content [24,25].

GX PCoVs were first reported by Lam et al. [26] in February 2020 from analysis of frozen tissue samples collected in Guangxi province between 2017 and 2018. The GX PCoVs form a separate clade to GD PCoVs and are more distantly related to SARS-CoV-2 [24,26–28]. The spike proteins of GX PCoVs have a higher amino acid similarity in the S1 N-Terminal Domain (NTD) to SARS-CoV-2 than GD PCoVs, but conversely a lower similarity in the RBD to SARS-CoV-2 and GD PCoVs (Supplementary Figures S1 and S2). Only two other discoveries of GX PCoV related sequences have been documented, one in frozen tissue samples sampled from a single pangolin in 2017 in Yunnan [29] and the other in 7 pangolin samples collected in 2018 in Vietnam [30]. GX PCoV sequencing data, however, is limited and of low quality and as such the potential for microbial profiling for viral host identification, and potential contamination is limited [30].

To identify viruses representing a high risk of crossover to humans, including a search for potential reservoirs of SARS-CoVs and SARS2r-CoVs, He et al. undertook metatranscriptomic analysis of 1725 game animals from 16 species across China [31]. The study included 423 *Paguma larvata* (Masked palm civet) specimens, a species implicated in bidirectional exchange of SARS-CoV with humans [32]. Both *Manis javanica* (Malayan) and *Manis pentadactyla* (Chinese) pangolins were also sampled. Notably, of these 16 species, 7 species or related species were sold at the HSM in November 2019 [24]. He et al. note "no viruses closely related to either SARS-CoV or SARS-CoV-2 (or other sarbecoviruses) were detected in any of animals examined", and no detection of SARS-CoV-related coronaviruses (SARSr-CoVs) is described in the virus species classification of the samples. However, Jones et al. unexpectedly identified a novel SARS2r-CoV in 6 pangolin (Malayan and Chinese) and 3 Malayan porcupine (*Hystrix brachyura*) metatranscriptomic datasets [24]. The SARS2r-CoV partial genome has high homology to the GX PCoVs in the NSP 4 N- and C-terminal regions but is more closely related to bat-SL-CoVZC45 in the RdRp coding region.

Here, we expand on our previous analysis to find GX_ZC45r-CoV, a novel bat-SL-CoVZC45-related coronavirus, in two coypu (*Myocastor coypus*), two Malayan porcupine, one each of Asian badger (*Meles leucurus*), Masked palm civet and Hoary bamboo rat (*Rhizomys pruinosus*) datasets sequenced by He et al., in addition to the six pangolin and three Malayan porcupine datasets described in Jones et al. [24]. We undertake more extensive mitochondrial alignments, human mitochondrial haplogroup analysis and more detailed phylogenetic analysis of genomic regions covered by the recovered sequences and propose that the fragmentary nature of the genome indicates genetic manipulation.

## 2. Methods

All SRAs were trimmed using TrimGalore v0.6.7 (https://www.bioinformatics.babraham.ac.uk/projects/trim_galore/) accessed on 10 March 2022 using default adapter detection. All

local blastn searches were made against a local copy of the NCBI nt database [33] downloaded on the 21 January 2021. Multiple sequence alignment was conducted using MUSCLE [34] in UGENE v42.0 [35].

## 2.1. Consensus Genome

For all datasets with SARS2r-CoV reads, we pooled trimmed forward and reverse paired end reads and single end reads separately. We aligned pooled paired- and single-end reads separately to both bat-SL-CoVZC45 and pangolin CoV GX_P4L using minimap2 using the following command line parameters "-MD -c -eqx -x sr -2 -t 20 –sam-hit-only –secondary=no". Paired- and single-end alignments were pooled for each set of bat-SL-CoVZC45 and PCoV GX_P4L aligned reads. Consensus genomes were generated using Samtools v1.15.1 [36] using default settings. The PCoV GX_P4L aligned consensus genome was used for the NSP4 region, while the consensus from alignment of reads to bat-SL-CoVZC45 was used for the NSP10 and RdRp regions (Supplementary Data: GX_ZC45r-CoV.fa). To generate a gap-filled genome, the NSP4 and NSP10 + RdRp consensus genomic regions were aligned to bat-SL-CoVZC45 using MUSCLE in UGENE and bat-SL-CoVZC45 was replaced by GX_ZC45r-CoV sequences in these two regions (Supplementary Data: GX_ZC45-CoV_ZC45_gap_filled_no_polyA.fa).

## 2.2. Viral Alignments

158 SRAs in PRJNA and PRJNA795267 were aligned to a set of viruses identified using fastv [37] and to GX PCoV using bowtie2 (http://bowtie-bio.sourceforge.net/bowtie2/index.shtml) accessed on 20 April 2021 using the '–very-sensitive' alignment option. Where GX PCoV was identified, mimimap2 [38] was to align the dataset to a 'GX_ZC45r-CoV gap-filled' genome reference using minimap2 version 2.24 with the following parameters "-MD -c -eqx -x sr –sam-hit-only –secondary=no -t 32".

All SRAs with GX_ZC45r-CoV sequences were pooled and aligned to bat-SL-CoVZC45 (MG772933.1) with the poly(A) tail removed and PCoV GX_P4L (MT040333.1) using bwa-mem version 0.7.17 [39] with default parameters.

Viral reference sequences were downloaded from NCBI (https://ftp.ncbi.nlm.nih.gov/refseq/release/viral/) on 11 May 2021. 64 SRAs from PRJNA793740 and PRJNA795267 (Supplementary Figure S18) were de novo assembled using MEGAHIT v.1.2.9 [40]. Final contigs were aligned to the viral reference set using minimap2 v2.22-r1101 with the following command line parameters -MD -c -eqx -2 -t 32 –sam-hit-only –secondary=no. Coverage was calculated using bamdst v1.0.9 (https://github.com/shiquan/bamdst) accessed on 5 July 2021. Results were plotted using matplotlib v3.3.4 (https://matplotlib.org/) accessed on 15 September 2021.

## 2.3. Phylogenetic Analyses

For phylogenetic analysis the following workflow was used: 100 SARSr-CoV genomes with highest Blastn percentage identity to each of the assembled NSP10 + RdRp region, NSP4 region and bat-SL-CoVZC45 were de-duplicated. The GX_WIV [28] genome sequence was added to the set. The sequences were then aligned using the MUSCLE algorithm in UGENE with default settings. Alignments were then manually trimmed to relevant sub-regions. For the 407 nt partial RdRp analysis, a 440 nt partial RdRp region of GX_ZC45r-CoV was aligned to 91 SARSr-CoV partial RdRp regions sampled by Hu et al., SARS-CoV-2 Wuhan-Hu-1, PCoV MP789, PCoV GX_P4L and BtKY72 also using MUSCLE in UGENE. The 3′ and 5′ ends of the combined alignment were trimmed leaving a 407 nt region. MUSCLE aligned genome sets were then analyzed using the model selection tool in MEGA11 [41].

The partial RdRp section (297 nt) maximum likelihood tree was generated using raxmlGUI 2.0 v2.0.8 [42] using the following parameters: –all –model GTR+I+G –seed 470,171 –bs-metric tbe –tree rand{1} –bs-trees 1000. All phylogenetic trees analysis was repeated using PhyML 3.0 [43] with automatic model section by smart model selection [44]. Default settings were used. Trees were exported and plotted in MEGA11.

*2.4. SimPlot Analyses*

*SimPlot++ groups for GX_ZC45r query plot, genome as named except:*
*ZXC21: bat-SL-CoVZXC21, ZC45: bat-SL-CoVZC45, PCoV_GX: PCoV_GX-P4L, PCoV_GD:*
*PCoV_MP789, HKU3: HKU3-1, FJ2021: FJ2021D, AH2021: AH2021A.*
*SimPlot++ groups for PCoV GX (PCoV_GX: GX_P2V, PCoV_GX-P1E, PCoV_GX-P4L, PCoV_GX-*
*P5E, PCoV_GX-P5L) query plot, single genomes except for these groups: PCoV_GD: PCoV_A22-2,*
*PCoV_MP789, PCoV_SM44-9, PCoV_SM79-9, BANAL: BANAL-20-103/Laos/2020, BANAL-20-*
*116/Laos/2020, BANAL-20-236/Laos/2020, BANAL-20-236/Laos/2020, BANAL-20-247/Laos/2020,*
*BANAL-20-52/Laos/2020.*
*SimPlot++_groups for PCoV GD (PCoV_GD: PCoV_A22-2, PCoV_MP789, PCoV_SM44-9,*
*PCoV_SM79-9) query plot, single genomes except for these groups: PCoV_GX: GX_P2V, PCoV_GX-*
*P1E, PCoV_GX-P4L, PCoV_GX-P5E, PCoV_GX-P5L, BANAL: BANAL-20-103/Laos/2020,*
*BANAL-20-116/Laos/2020, BANAL-20-236/Laos/2020, BANAL-20-236/Laos/2020, BANAL-20-*
*247/Laos/2020, BANAL-20-52/Laos/2020.*

## 3. Results

We undertook analysis of all SRA data in BioProjects PRJNA793740 and PRJNA795267 and expand upon [24] to identify additional GX_ZC45r-CoV sequences in two coypu, two Malayan porcupine and one in each of Hoary bamboo rat, Asian badger and Masked palm civet SRA datasets.

We aligned each SRA dataset to a reference genome 'GX_ZC45r-CoV gap-filled' which consisted of a GX_ZC45r-CoV partial genome generated as per methods, with missing regions replaced with bat-SL-CoVZC45 (MG772933.1). Coverage for each of the 16 game animal datasets is solely in the non-structural protein 4 (NSP4), non-structural protein 10 (NSP10) and RNA dependent RNA polymerase (RdRp) coding regions (Figure 2). The number of reads mapping to 'GX_ZC45r-CoV gap-filled' in the seven additional game animal samples was very low, at between 1 and 8 reads (Supplementary Figure S3).

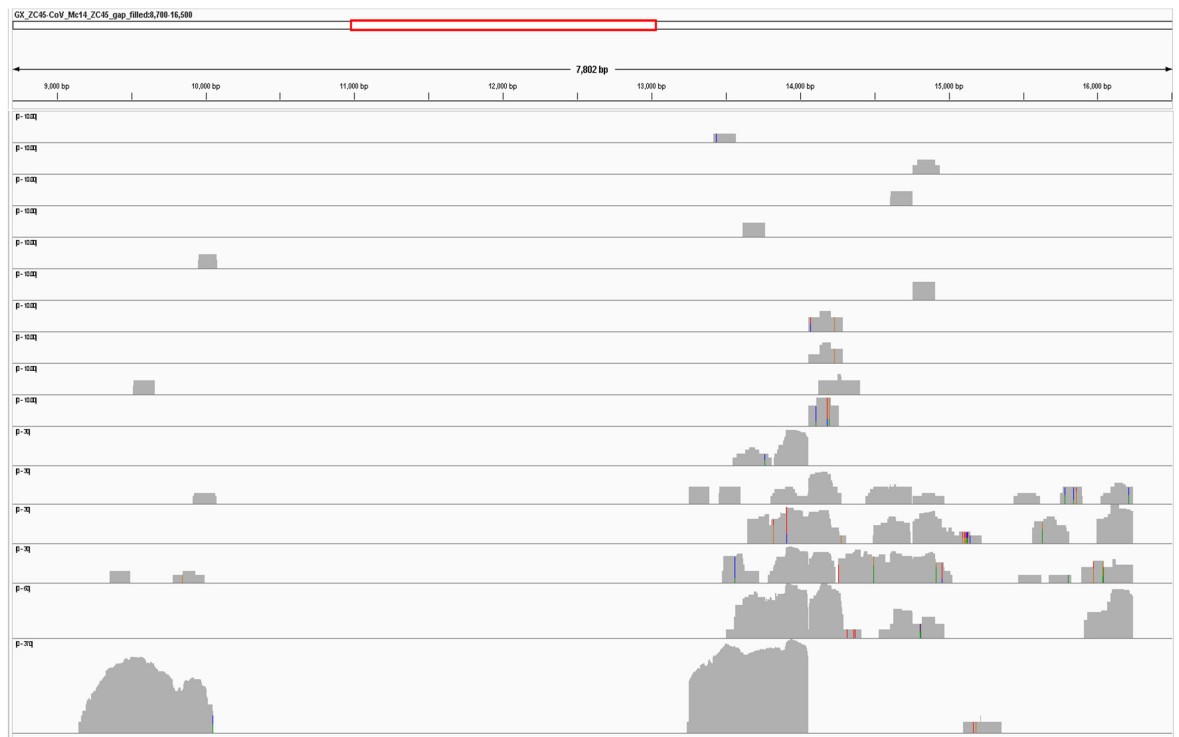

**Figure 2.** GX_ZC45r-CoV gap_filled genome coverage by sample (top to bottom tracks): PL-AH-MO-5, MC-HeB-T-1, HB-HuB-A-2, MJ-ZJ-MO-4, MJ-ZJ-MO-6, ML-HeB-F-1, HB-HuB-A-1, MC-HuN-T-1,

HB-FJ-NA-3, RP-JX-A-2 (0–10 range), HB-HuB-N-3, MJ-ZJ-MO-3, MJ-ZJ-MO-1, MP-ZJ-MO-4 (0–30 range), MJ-ZJ-MO-2 (0–60 range), HB-FJ-NA-7 (0–370 range). All tracks shown in log scale. Plotted using Integrative Genomics Viewer (IGV) [45].

Numerous SNVs relative to bat-SL-CoVZC45 are consistent across samples, with no consensus SNV absent in other samples, indicating the same strain is found in each of the samples.

Pooled reads from the 16 SRAs containing GX_ZC45r-CoV were aligned to PCoV GX_P4L and bat-SL-CoVZC45. 47 SNVs are found in the NSP4 region when aligned to PCoV GX_P4L (Figure 3). 104 SNVs and one 18nt missing section (15,421–15,438 nt) are found in the NSP10/RdRp region when aligned to bat-SL-CoVZC45 (Figure 4). Coverage of the NSP4 coding region is incomplete, with complete coverage of the 3′ end, including complete coverage of the NSP4 C-terminus (NSP4_C) but only 50% coverage of the NSP4 transmembrane domain (NSP4_TM).

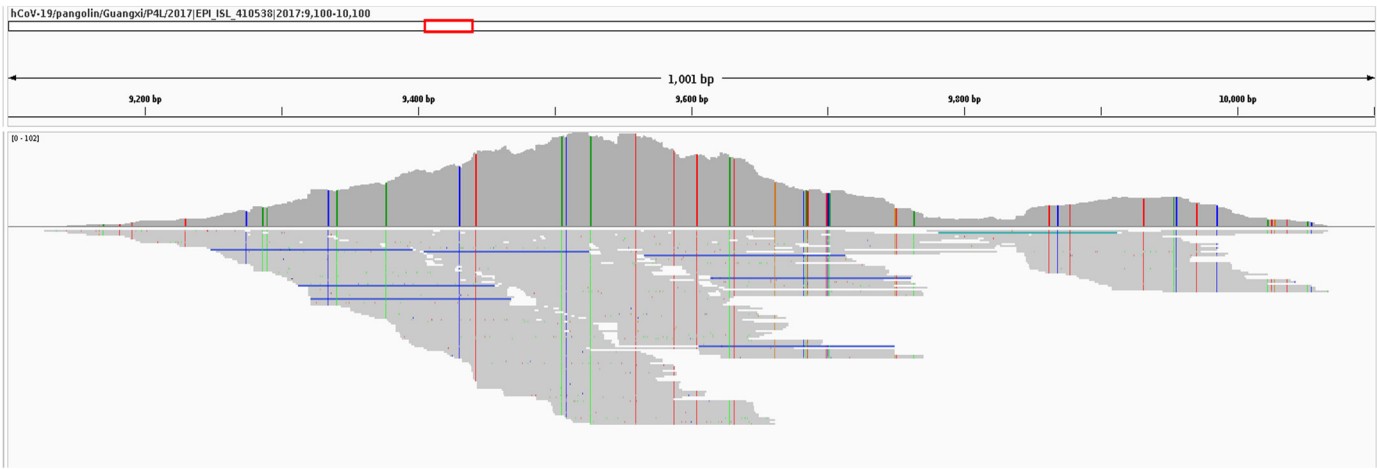

**Figure 3.** Alignment of pooled reads to PCoV GX_P4L, zoomed into the NSP4 region of ORF1a showing 952nt section of mapped reads. Reads coloured by strandness. SNV positions relative to PCoV GX_P4L are shown as vertical lines.

Coverage of the RdRp coding region is complete except for a 18 nt missing section, however coverage of the NSP10 coding region is incomplete with 47% of the 5′ end of NSP10 not covered by GX_ZC45r-CoV matching reads. We note that a 590 nt region at the 5′ end of the RdRp coding region (13,467–14,056 using bat-SL-CoVZC45 as a reference) has markedly higher read coverage than the rest of the RdRp, with an abrupt change at 14,057 nt. A second anomalous read coverage distribution occurs around position 14,758 nt relative to the bat-SL-CoVZC45 genome. These indicate the genome may have been sequenced in fragments, potentially as parts of a reverse genetics system [24] (Supplementary Figures S4 and S5).

### 3.1. Mitochondrial Mapping Analysis

The 16 datasets analyzed in Figure 2 were subjected to a systematic mitochondrial mapping analysis, which mapped the reads to all mitochondrial genomes present on NCBI (Supplementary Info. 1 and Source Code). The datasets are heavily contaminated with a range of unexpected mammalian and other eukaryotic species (Supplementary Figure S6). Presumably, if GX_ZC45r-CoV was derived from a cell line or animal tissue, then the relevant mitochondrial genome should be present in all the datasets where the viral sequences are detected. Common mammalian species present are *Homo sapiens* (ranging from 16 to 98% mitochondrial genome coverage in all datasets), and *Mus musculus* (ranging from 13 to 71% genome coverage in 13 datasets, and <10% in MJ-ZJ-MO-1, PL-AH-MO-5 and ML-HeB-F-1).

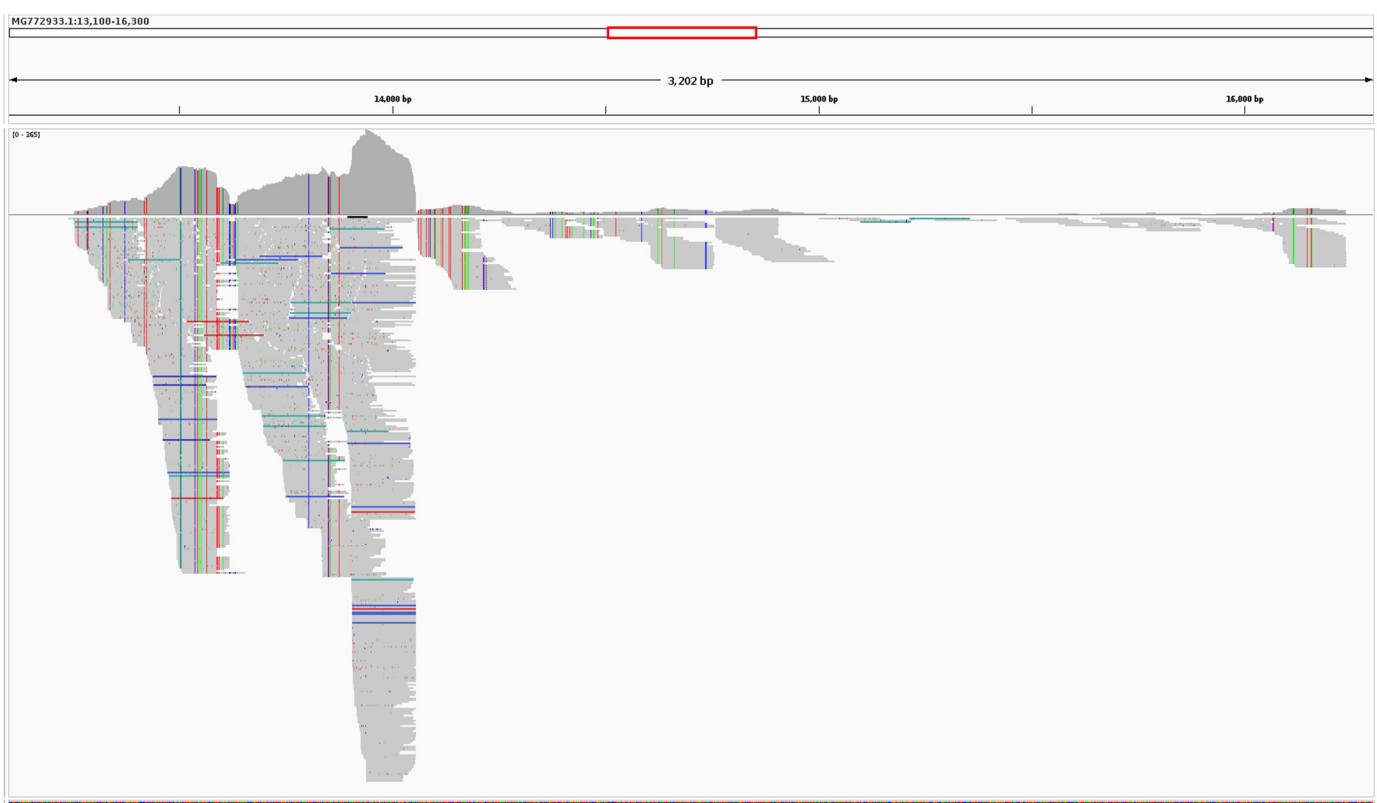

**Figure 4.** Alignment of reads to bat-SL-CoVZC45, zoomed in to the NSP10 and RdRp regions. Reads coloured by strandness. SNV positions relative to bat-SL-CoVZC45 shown as vertical lines.

Interestingly, both *Homo sapiens* and Masked palm civet mitochondrial genome coverage are higher than the Asian badger mitochondrial genome in the Asian badger dataset ML-HeB-F-1. The mitochondrial genome for *Laodelphax striatellus,* an insect vector for Rice stripe virus [46], and *Malassezia restricta,* a human hosted basidiomycetous yeast, are both found with high coverage in all samples except ML-HeB-F-1 and PL-AH-MO-5. However, environmental *Malassezia* spp. with DNA similar to *M. restricta* may be widespread in the environment [47]. *Debaryomyces fabri*, a salt tolerant yeast found in varied environments [48], including human skin and is also common to 8 of the 16 datasets. Several species of Aspergillus and Candida yeasts are also found. The widespread yeast occurrences may be indicative of cell culture with yeast growth contaminating the samples, contamination during incubation [49], or yeast contamination of materials during the library preparation stage.

### 3.2. Identification of Human Mitochondrial Haplogroups

Reads that mapped to the human mitochondrial genome were used to infer the human mitochondrial haplogroup in the 14 datasets that had significant human mitochondrial genome coverage. The datasets were mapped to the rCRS human reference mitochondrial genome (Accession J01415.2) using bowtie2 [39]. Then, mixemt [50] was used to infer the mitochondrial haplogroup. The results are displayed in Supplementary Table S1.

A dominant haplogroup, F1c1(a1) was identified in 12 of the 14 datasets. Of the remaining two datasets, MJ-ZJ-MO-3 did not possess any reads that mapped to rCRS, while in MJ-ZJ-MO-2 100% of human mitochondrial reads were attributed to the H27/H27e haplogroup. A number of minor haplogroups were observed in the remaining datasets: H1aw and H1t2 (MC-HuN-T-1), C (MC-HeB-T-1) and H27/H27e (MJ-ZJ-MO-4).

Haplogroup F1c1a1 is of East Asian origin, and its presence appears consistent with either worker contamination, or a human cell line of East Asian provenance. Haplogroup H27/H27e is of European/Central Asian origin, haplogroup C is found in Northeast Asia and the Americas, while haplogroup H1 is found in Europe and North Africa. Given

their geographical origin, haplogroups H27/H27e and H1 are inconsistent with worker contamination, and could represent cell lines. Given that haplogroup F1c1(a1) dominates the datasets, then if GX_ZC45r-CoV were associated with a human cell line, this haplogroup would be the likeliest candidate. Alternatively, if haplogroup F1c1(a1) derived from worker contamination then this indicates upstream contamination of sequencing reagents/samples rather than index hopping during sequencing. This observation may be useful when considering the source of the GX_ZC45r-CoV contamination.

### 3.3. Simplot Analysis

Similarity plot analysis was conducted using SimPlot++ [51] to review the two recovered sections of the GX_ZC45r-CoV genome (Figure 5). The entire recovered section of the NSP4 coding region has the highest similarity to GX PCoVs. The short recovered section of the NSP10 region (188 nt when a 19 nt section of the non-coding region between NSP10 and the RdRp coding region is included) shows several genomes to have high similarity, including bat CoV (BtCoV) Longquan-140 and bat-SL-CoVZC45.

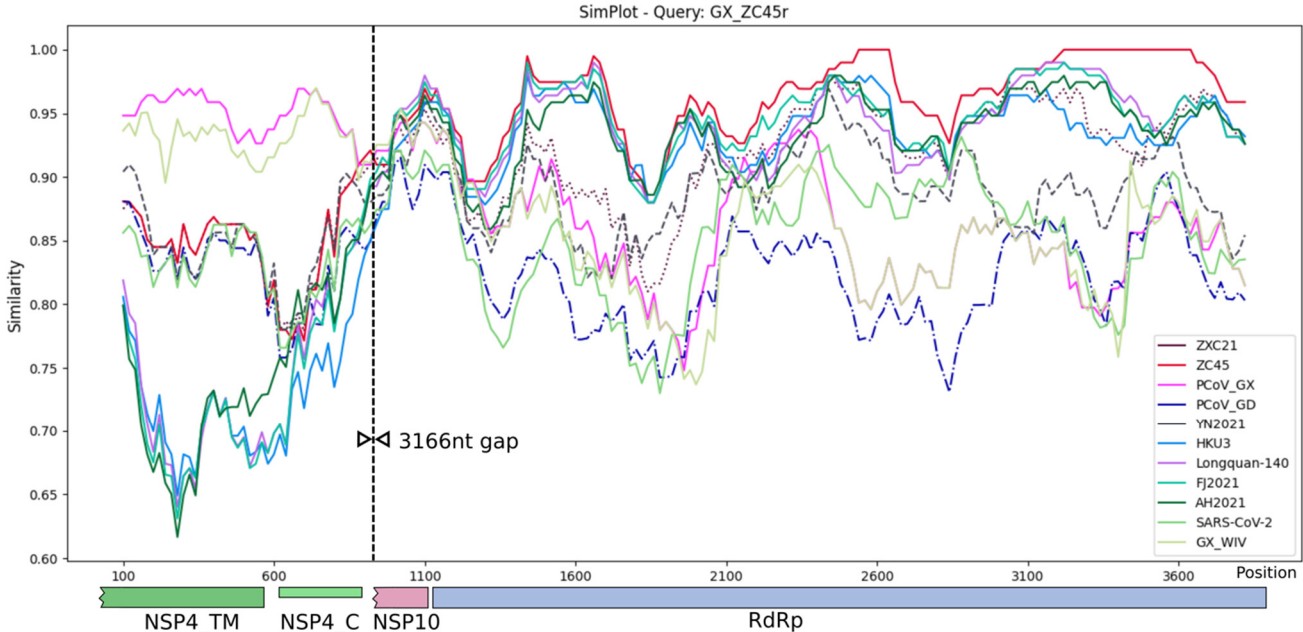

**Figure 5.** Simplot analysis of NSP4 region spliced with NSP10 and RdRp gene regions. Plotted using Simplot++ using a 200 bp window, 20 bp step and Kimura (2-parameter) distance model. A 3166 nt section (relative to multi-genome alignment) between covered regions was removed prior to analysis. Solid lines were used except: bat-SL-CoVZC21—dotted, Sarbecovirus sp. isolate YN2021—dashed, GD PCoV—dash-dot. See Methods for virus groupings.

Blastn analysis for the same 188 nt NSP10 region (including a 19 nt section of the 5′ end of the non-coding region between NSP10 and the RdRp coding region) shows *Sarbecovirus* sp. isolate FJ2021D to have highest identity at 96%, which includes 6 SNVs, one deletion and one insertion relative to GX_ZC45r-CoV. However, PCoVs GX_P5E, GX_P5L, GX_P1E, GX_P4L, BtCoV Longquan-140, and BtCoV SC2018B all have 9 SNVs for 95% identity, for a more parsimonious match not requiring a deletion and insertion. Interestingly, although the absolute number of nucleotide differences is the same or very similar between the bat CoVs and PCoVs, the positions of the differences are almost all completely different between the two groups (Supplementary Figure S7).

Over the RdRp coding region, the highest identity over almost the entire region is to bat-SL-CoVZC45, with two regions of 100% identity. However, distinct regions of <95% identity to bat-SL-CoVZC45 are evident, indicating significant evolutionary distance from bat-SL-CoVZC45.

GD PCoV MP789 and the GX PCoV group were also analyzed using SimPlot++ and queried against SARSr-CoVs with high identity over at least part of the GX_ZC45r-CoV partial genome. PCoV MP789 exhibits high identity to bat-SL-CoVZC45, bat-SL-CoVZXC21, RacS271 and RaTG13 in the NSP4 region, but low identity to bat-SL-CoVZC45 and bat-SL-CoVZXC21 over the RdRp coding region (Supplementary Figure S8). The GX PCoV group exhibits a distinctly high identity to GX_ZC45r-CoV in the NSP4 coding region (Supplementary Figure S9). However, over the RdRp coding region of the genomes analyzed, the GX PCoV group only exhibits highest identity to GX_ZC45r-CoV over a 297 nt region (2144–2440 nt in Supplementary Figure S9). To quantify the match blastn was used with the 297 nt section of GX_ZC45r-CoV as the query and PCoV GX_P4L as the subject and a 93.94% identity was found (279/297 nt match). Blastn was again used to analyze this region of PCoV GX_P4L, which was located at position 14,432–14,728, which resulted in the highest identity to any genome on NCBI of 91.84% (SARS-CoV-2, Accession OU470778.1). This confirms that GX_ZC45r-Cov is the closest known match in this part of the RdRp gene.

*3.4. Phylogenetic Analysis*

Phylogenetic trees were constructed for the NSP4, NSP10, RdRp and partial RdRp coding regions which were covered by GX_ZC45r-CoV reads. For the NSP4 region, which plays a role in assembling the viral double membrane vesicles, a GTR+G+I model was estimated as having the lowest Bayesian information criterion (BIC) and 5 discrete gamma categories were used. GX_ZC45r-CoV exhibits a basal sister relationship to GX CoVs with unanimous support. GX_WIV [28] exhibits a more divergent sequence in this region than related GX CoVs (Figure 6). A maximum likelihood tree implemented using PhyML using a GTR+G+I model shows the same basal sister relationship of GX_ZC45r-CoV to GX PCoVs (Supplementary Figure S24a).

Phylogenetic analysis of the NSP10 region, which plays a role in mRNA cap methylation, shows GX_ZC45r-CoV to have a basal sister relationship to the GX PCoV clade (Figure 7). GD PCoVs however are more closely related to the SARS-CoV-2/BANAL clade and form a basal sister clade. Bat-SL-CoV-ZC45 is significantly more divergent from the PCoV GX_PD and SARS2r CoV clades, and is located in the SARSr-CoV clade. Repeat analysis using PhyML with a TN93+G model shows the same relationships (Supplementary Figure S24b).

In a maximum likelihood tree covering the RdRp coding region, bat-SL-CoVZC45 and GX_ZC45r-CoV form a clade with an ancestral node in common with HKU3-1 and BtCoV Longquan-140 and sits on the SARSr-CoV branch (Figure 8). Similar to the NSP10 phylogenetic tree, all GD PCoVs form a sister clade to the SARS-CoV-2/BANAL clade. A maximum likelihood tree using PhyML with a GTR+G model also shows GX_ZC45r-CoV on the same branch as bat-SL-CoVZC45 (Supplementary Figure S24c). A blastn analysis of the RdRp of GX_ZC45r-CoV (including 18 nt missing nucleotides) confirms ZC45 has closest identity at 95.85%.

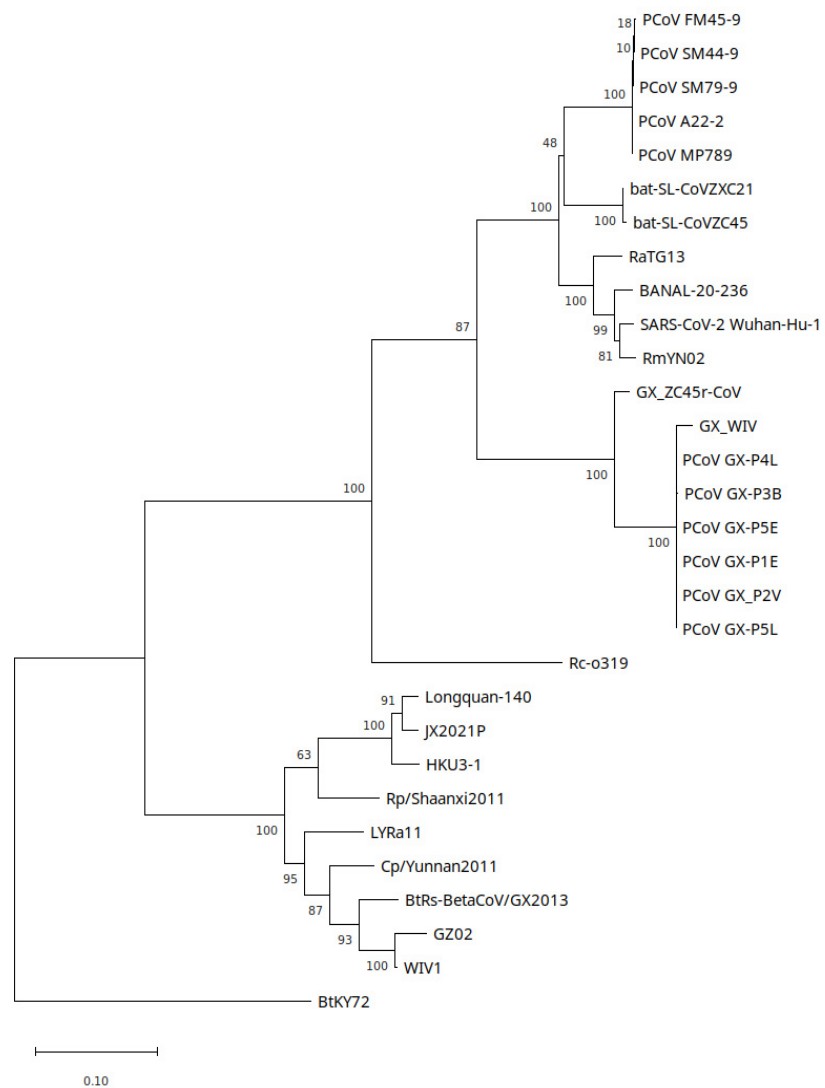

**Figure 6.** Partial NSP4 region maximum likelihood tree constructed using a GTR+G+I model with 1000 bootstrap replicates. Branch support percentage is shown next to the branches. The tree is drawn to scale, with branch lengths measured in the number of substitutions per site. Several genomes only had partial coverage of this region: GX_WIV 77.8%; PCoV_GX_P3B 77%; PCoV_FM45-9 98.65%. MP20 had only 11.1% coverage and was excluded from analysis.

The phylogenetic relationship of PCoV MP20 to the GX PCoVs in the RdRp gene is similar to the relative relationship of GX_ZC45r-CoV to the GX PCoVs in the NSP4 and NSP10 regions. To further compare the similarity of PCoVMP20 to GX_ZC45r-CoV, a similarity plot was constructed using SimPlot++ using PCoV MP20 as the query against selected SARS2-CoV genomes. As coverage of both GX_ZC45r-CoV and PCoV MP20 is limited, the four regions over which both genomes had coverage were spliced to form a single contiguous pseudo genome for analysis (Supplementary Figure S10). Where data is available, the two genomes exhibit moderate identity, ranging between 70–90% except for an approximately 40 nt section of the RdRp coding region with a 95–97% identity.

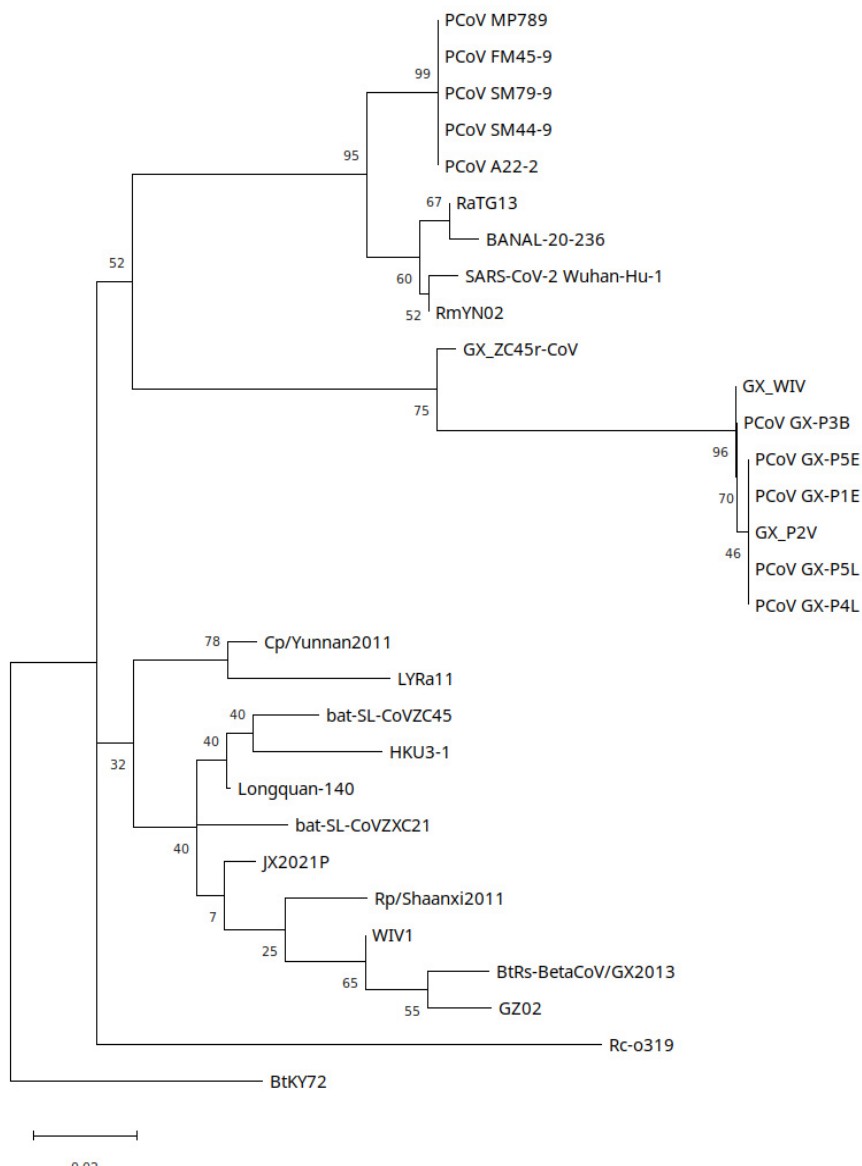

**Figure 7.** NSP10 region maximum likelihood tree using Kimura 2-parameter model (K2+G) with500 bootstrap replications. 5 discrete gamma categories. Branch support percentage is shown next to the branches. The tree is drawn to scale, with branch lengths measured in the number of substitutions per site.

A maximum likelihood tree was generated to test the phylogenetic relationship of a 297 nt section of the RdRp gene of PCoVGX_P4L which was found to have highest blastn identity to GX_ZC45r-CoV (Figure 9). Model selection in MEGA11 was used to identify a T92+G model to have lowest BIC and a GTR+G+I model to have lowest Akaike information criterion (corrected for small same size) (AICc) score, and was found to generate a phylogenetic reconstruction with higher branch support. Similar to findings for the NSP4 and NSP10 coding regions, GX_ZC45r-CoV is situated in a basal sister relationship to GX PCoVs, but with poor branch support. However, maximum likelihood trees using raxmlGUI with a GTR+I+G model and PhyML using a HKY85+G model place GX_ZC45r-CoV in a basal position to GX PCoV and SARS-CoV-2/GD PCoV clades (Supplementary Figures S11 and S25a).

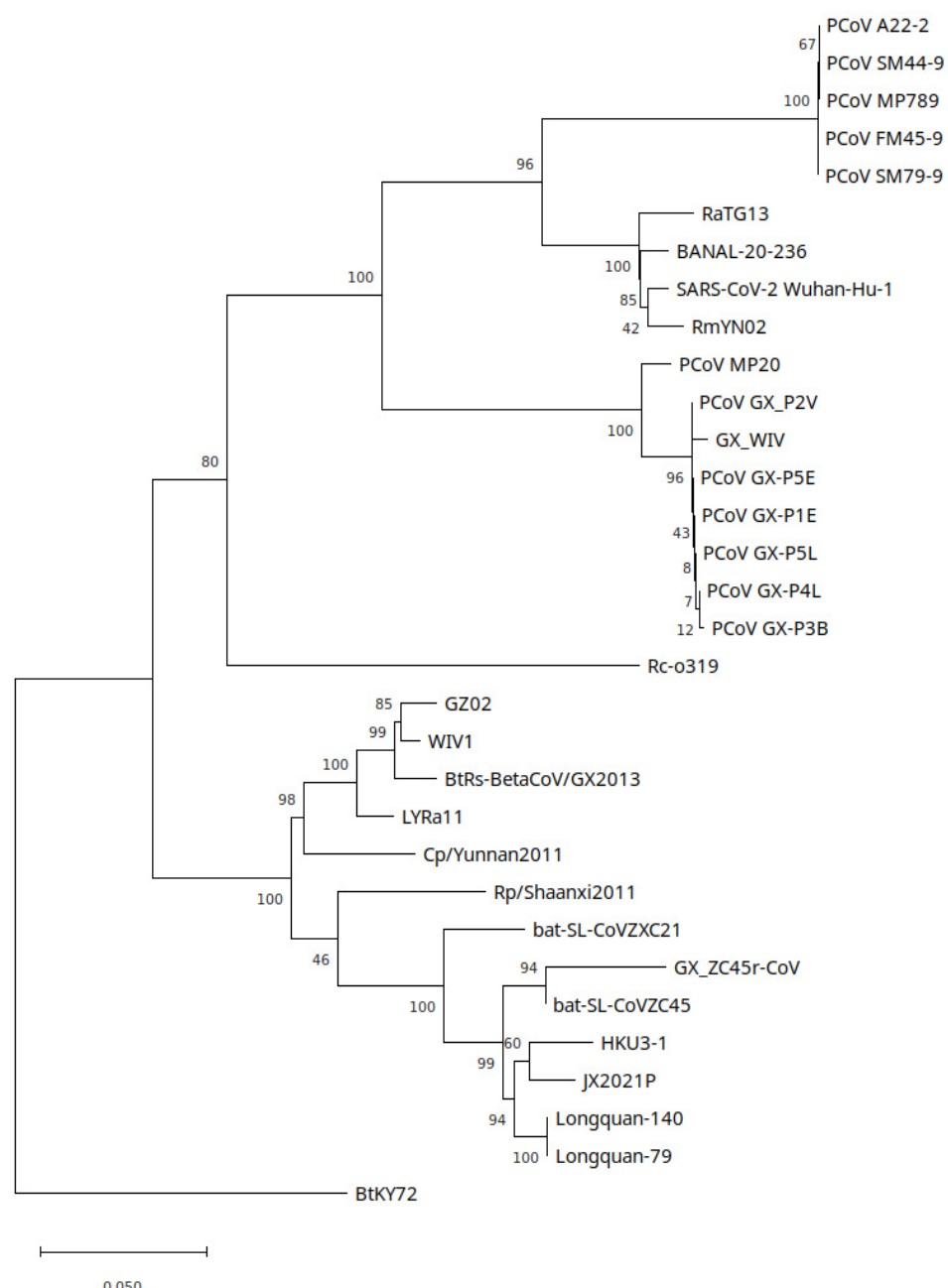

**Figure 8.** RdRp region (2795 nt) maximum likelihood phylogenetic tree using GTR+G+I model
with 1000 bootstrap replicates. Branch support percentage is shown next to the branches. The
tree is drawn to scale, with branch lengths measured in the number of substitutions per site. The
following genomes have partial coverages: GX_ZC45r-CoV, GX_WIV, PCoV_GX-P3B, PCoV MP20,
PCoV_FM45-9, comprising 99.4%, 96.5%, 93.6%, 42.6%, and 98.7% respectively.

Hu et al. sampled 334 Zhoushan island bats and identified 89 to be carrying SARSr-
CoVs, however only two full genomes were recovered, bat-SL-CoVZXC21 and bat-SL-
CoVZC45 [13]. 89 partial RdRp amplicon sequences were submitted to GenBank (Acces-
sions MG772844 through MG772932). We aligned this set of partial sequences, together
with GX_ZC45r-CoV and selected genomes and trimmed the multiple sequence align-
ment to 407nt. Unfortunately, the 440 nt PCR targeted section of the RdRp covered the
18 nt gap and a low read depth section of GX-ZC45r-CoV. bat-SL-CoVZC45 had lowest
Hamming dissimilarity, with 3 nt SNVs relative to GX-ZC45r-CoV. 11 other Zhoushan
bat CoVs also exhibited low dissimilarity with 4 or 5 SNVs in this region. A maximum

likelihood phylogenetic tree was constructed using the 23 partial RdRp sequences from Hu et al. [13] with lowest Hamming dissimilarity to GX_ZC45r-CoV, and selected other genomes (Supplementary Figure S12). A maximum likelihood tree using PhyML with a T93+G model generated a very similar result (Supplementary Figure S25b). In general, the selected partial Zhoushan RdRp sequences exhibit relatively low differentiation, with GX_ZC45r-CoV clustering within this clade.

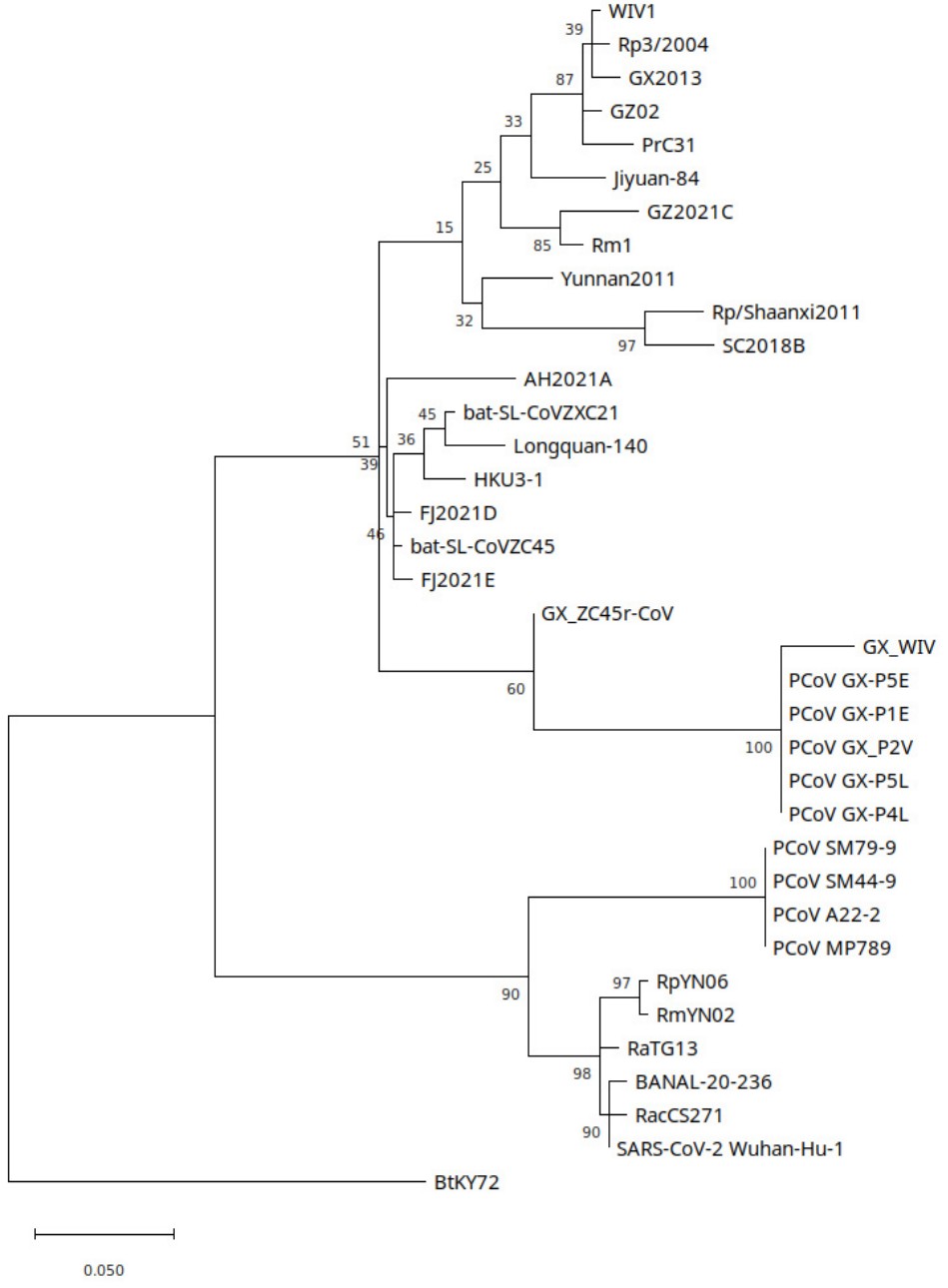

**Figure 9.** Partial RdRp section (297 nt) maximum likelihood phylogenetic tree using a GTR+G+I model with 100 bootstrap replicates. 7 discrete Gamma categories were used. Generated using MEGA11.

*3.5. Recombination Analysis*

As the phylogenetic relationships for different parts of GX_ZC45r-CoV differ significantly, we undertook recombination analysis to determine if recombination breakpoints could be identified. We used RDP5 [52] which implements the RDP [53], GENECONV [54], Chimaera [55], MaxChi [56], BootScan [57] and SiScan [58] methods. Two potential recombi-

nation regions were detected by more than three methods (Supplementary Table S2). The 3′ breakpoint of recombination region 1 was detected at approximately the NSP4/NSP10 splice location where a 3166 nt gap in genome coverage was removed from alignments, while the 5′ end of this potential recombination event was not detected (Supplementary Figure S13). However, although possible recombination region 1 is wholly located within fragment 5 of Temmam et al. [58], the detected breakpoint could be artifactual given that it coincides with a missing section of the genome. Greater genome coverage is required to place higher confidence on, and identify breakpoint location for this potential recombination region.

Potential recombination region 2 is located within the RdRp coding region (Supplementary Figure S14). The major parent identified was bat-SL-CoVZC45 while the minor parent was not identified from the genomes used in the alignment (Supplementary Table S2). The recombination breakpoints for recombination region 2 were not identified in any of the other sequences analyzed. Neither (possible) recombination regions 1 or 2 were previously identified in other SARS2r-CoV genomes [14,59,60].

Complicating potential recombination analysis, as GX_ZC45r-CoV appears to have been sequenced as cDNA in plasmids, it is also possible that recombination could have been introduced artificially. However, two possible artificial splices indicated by read depth pattern were not detected as breakpoint sites using RDP5 and inspection of the multiple sequence alignment does not indicate an obvious genome change at these positions.

*3.6. Synthetic Vectors*

Using a custom python script (Source Code) we searched for common primer and promoter sequences in 23 de novo assembled SRA datasets in PRJNA793740 and 15 de novo assembled datasets in PRJNA795267, including all SRAs with GX_ZC45r-CoV sequences. Contigs detected with vector sequences were reviewed using Addgene sequence analyzer (https://www.addgene.org/analyze-sequence/) accessed on March 2022 and Blastn. Fragments of synthetic vectors were detected in all datasets. A partial vector containing an SV40 promoter and Neomycin selection marker [61] was found in MJ-ZJ-NA-2 and MJ-ZJ-F-1 and MH-ZJ-F-1, with the SV40 promoter sequence also found in the HB-FJ-L-2 dataset (Supplementary Figure S15). The partial vector sequence has a similar layout to pSV2 neo, a plasmid used for mammalian cell line expression [62]. A partial vector containing a 575 nt sequence of the Woodchuck Hepatitis Virus Posttranscriptional Regulatory Element (WPRE) and puromycin resistance gene (puro) is found in sample MC-GX-A-1 (Supplementary Figure S16), with shorter partial vectors containing WPRE found in MJ-ZJ-MO-1, MP-ZJ-MO-4 and MC-GX-F-1. WPRE is widely used in viral vectors to increase viral expression and titres [63].

Highest coverage of a Tn5 transposase vector (4275-4570nt) was identified in 6 datasets (MJ-ZJ-NA-2, MJ-ZJ-F-1, MJ-ZJ-F-3, MH-ZJ-F-1, MJ-ZJ-F-6 and HB-FJ-F-4) with shorter sequences (478-3,738 nt) in 7 datasets (MJ-ZJ-MO-6, MJ-ZJ-MO-3, MC-HeB-T-1, MC-GX-A-1, MJ-ZJ-MO-1, MC-HuN-T-1 and HB-HuB-N-3) (Supplementary Figure S17). Contamination by library construction kit materials [64,65] may have been the source of this vector.

*3.7. Human and Mouse Hosted Viruses*

We used NCBI STAT to analyze the taxonomy of all 239 SRA datasets in BioProjects PRJNA793740 and PRJNA795267. We additionally ran fastv analysis against the Opengene viral genome kmer collection on 186 of the SRA datasets. We selected viruses that were commonly found across the SRA datasets and aligned 158 datasets to a reference set of viruses using bowtie2. We selected 46 datasets including all those with GX_ZC45r-CoV, Human orthorubaulavirus 2, and Woodchuck Hepatitis Virus sequences for analysis (Figure 10, Supplementary Figures S18–S20).

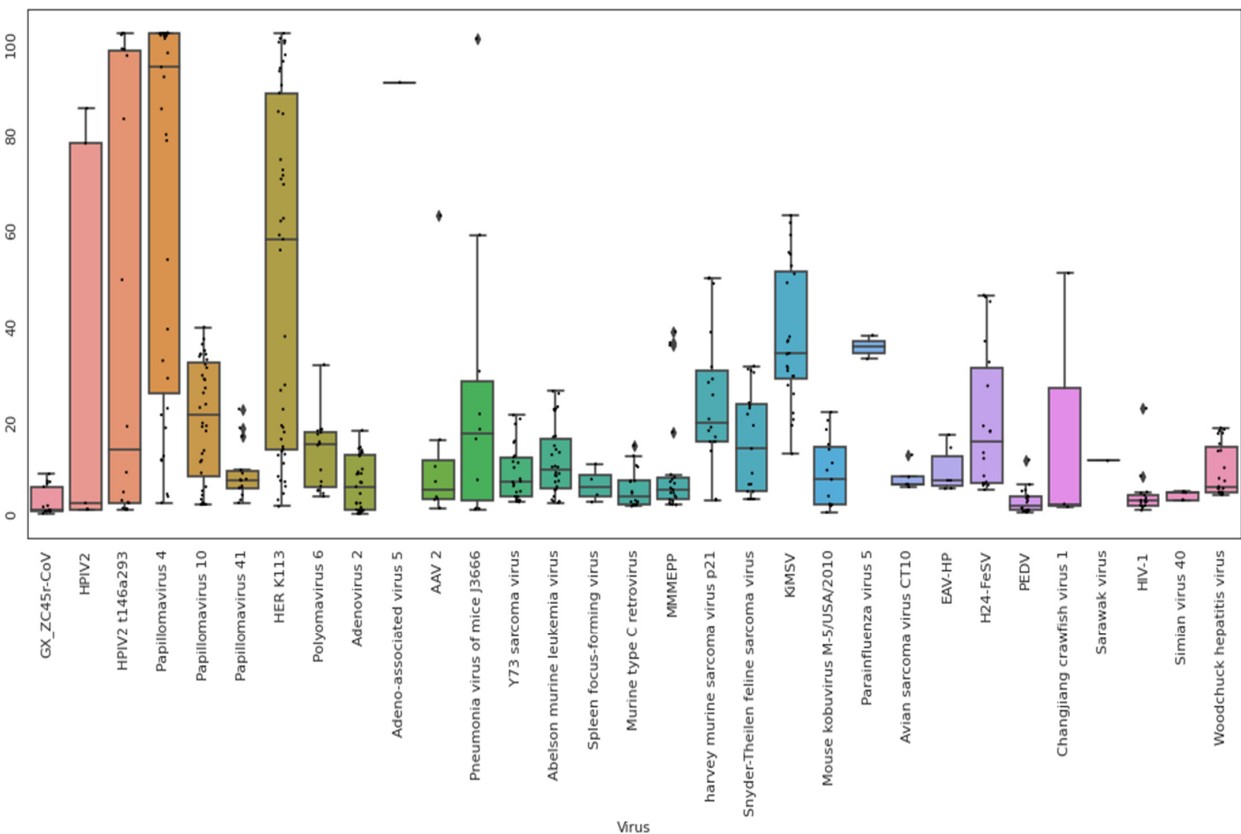

**Figure 10.** Box plot of percent coverage of selected viruses for selected SRAs in BioProjects PR-JNA793740 and PRJNA795267. Viruses were included only when each virus was present in at least one of the 46 datasets with 10% or greater genome coverage. However, a cutoff was not applied to GX_ZC45r-CoV and SV40. See Supplementary Info. 2 for virus descriptions.

Human hosted viruses are found in high count numbers and coverage including Human Papillomaviruses (HPVs) and Human Endogenous Retrovirus (HERV) K113. Human Orthorubulavirus 2 (Human parainfluenza virus 2, HPIV2)) strain t146a293_HPIV2 (Accession MH892406.1) is found at extremely high levels in pangolin samples MJ-ZJ-MO-4 and MJ-ZJ-MO-3 at 5.91% and 27.66% of the metagenome, respectively, and the same strain was identified in *Marmota himalayana* (Himalayan marmot) sample MH-HeB-NA-1 [24]. Using a Spearman correlation matrix, the presence of HPIV2 is moderately correlated with HPV and HERV K113 abundances (Supplementary Figure S21). Although He et al. proposed that the HPIV2 was a novel strain naturally infecting Malayan pangolins [31], the presence of HPIV2 in samples from 5 animal species (Supplementary Figure S18) and extremely high levels in two samples shows the presence of HPIV2 is very likely related to upstream contamination of the datasets and not related to natural infection of the sampled animals. It is interesting to note that of the 16 datasets identified in the BioProjects sequenced by He et al. that contain GX_ZC45r-CoV, 8 also contain HPIV2 indicating a possible causal link between the presence of GX_ZC45r-CoV and the presence of HPIV2.

To further determine potential contaminating viruses, de novo assembled contigs from 64 SRAs were aligned against the NCBI complete viral RefSeq set (Supplementary Figure S22). Similar to read alignment results, a correlation is evident between the presence of GX_ZC45r-CoV sequences and high coverage of HPV4, HPV10, HERV K113 and fair correlation with HPIV2.

Numerous mouse hosted viruses are also present in both read and contig alignments, consistent with mouse sequence contamination discussed above. In read analysis, pneumonia virus of mice J3666 is found in highest abundance and coverage in pangolin samples MJ-ZJ-MO-3, MJ-ZJ-MO-6, MJ-ZJ-MO-4, MJ-ZJ-MO-2 and MP-ZJ-MO-4.

Kirsten Murine Sarcoma Virus (KiMSV) is found with significant genome coverage in multiple Malayan porcupine, Hoary bamboo rat, coypu and Malayan pangolin samples (Supplementary Figure S20).

## 4. Discussion

Our findings are consistent with Jones et al. that the GX_ZC45r-CoV sequences identified in multiple game animal datasets, are contamination related [24]. Here, we identified a small number of reads mapping to GX_ZC45r-CoV in seven additional metatranscriptomic datasets sequenced by He et al. [31]: two coypu, two Malayan porcupine, and one each of Asian badger, Masked palm civet and Hoary bamboo rat datasets [31]. The additional CoV reads map to the same three regions identified by Jones et al. [24]: the NSP4, NSP10 and RdRp coding regions.

The non-random mapping of NGS reads to the GX_ZC45r-CoV gap-filled reference genome, solely to the NSP4, NSP10 and RdRp coding regions, across multiple game animal datasets is suggestive of both a single contamination source, and genetic manipulation of the coronavirus. If the coronavirus was present in the originating samples as a wild-type virus or 'live' isolate then the reads should map across the majority of the reference genome. Bat-SL-CoVZC45 was isolated from *R.pusillus* bats [13]. However, no reads mapped to any *Rhinolophus* sp. mitochondrial genome in the 16 datasets examined in Figure 2, with the exception of MP-ZJ-MO4, which displays 0.88% coverage of both the *Rhinolophus rex* and *Rhinolophus thomasi* mitochondrial genomes, an amount which is too low to be meaningful (Supplementary Info. 1). While the possibility exists that an enrichment procedure was used to isolate the virus, this does not explain the fragmentary nature of the sequences. This observation supports the interpretation that the GX_ZC45r-CoV sequences are not wild-type, as they are not associated with *Rhinolophus* spp. reads. This implies it is either a 'live' isolate cultured in a non-Rhinolophid cell line, or represents some form of cloning experiment.

Interestingly, Hu et al. [13] infected live mice with bat-SL-CoVZC45 (reported as rat due to a mistranslation (Noted by Andre Goffinet on Twitter https://archive.ph/wip/kBowy) 8 April 2022. This indicates that bat-SL-CoVZC45 could be culturable in a mouse cell line. We note that all but three of the SRA datasets listed in Figure 2 have reads mapping to the mouse mitochondrial genome with coverage of the genome at >10%. Alternatively, the presence of human mitochondrial reads in all of the datasets presents the possibility that GX_ZC45r-CoV was being cultured in a human cell line. The majority of the reads that map to the human mitochondrial reference genome belong to haplogroup F1c1a1, which is of East Asian origin, and may represent worker contamination, or alternatively a cell line of East Asian provenance. However, the non-random mapping of reads to the GX_ZC45r-CoV gap-filled genome appears inconsistent with a 'live' viral culture.

In coronaviruses, infectious clones are typically synthesized as fragments, which are ligated and then inserted into a variety of expression systems [66]. Multiple methods are used including: in vitro ligation using type II restriction endonucleases [67,68] Vaccinia virus vectors [69] a bacterial artificial chromosome (BAC) system [66,70,71] and transformation associated recombination [72].

Virus rescue is conducted by transfection into a cell line that supports overexpression of the infectious clone RNA. The fragmentary nature of the mapping to the GX_ZC45r-CoV gap-filled genome might suggest that parts of an infectious clone are present in the datasets. This is further supported by the identification of a 21 nt pUC57 (multiple cloning site (MCS) sequence at the 5′ end of a single read comprising the NSP10/RdRp coding fragment. It is interesting to note that the pUC57 plasmid, a common *Escherichia coli* high copy number plasmid, is widely used for coronavirus reverse genetic systems [73]. For comparison, the SARSr-CoV WIV1 BAC infectious clone was split into 8 fragments of lengths from 1518 to 5451 nt [71]. However, the 5′ end of fragment D, which contained the NSP10 and RdRp coding regions, was located near the 3′ end of NSP8. In addition, fragment C1 which

contains the WIV1 NSP4 coding region is 2619 nt, significantly longer than the 952 nt long NSP4 coding section recovered in GX_ZC45r-CoV [71].

If indeed the reads that map to the GX_ZC45r-CoV genome (Figure 2) belong to components of an infectious clone then this implies that unpublished SARS2r-CoV genomes and reverse genetics systems are at present being studied in China. Unpublished beta-coronavirus reverse genetics systems are not unprecedented, however, as an unpublished HKU4r-CoV infectious clone with highest homology to BtTp-BetaCoV/GX2012 was identified in a HZAU sequenced agricultural dataset [74].

The variant positions revealed when the reads are mapped to the NSP10-RdRp region of the bat-SL-CoVZC45 genome (Figure 4) are interesting as they indicate some evolutionary divergence. This could represent an independent viral lineage, or divergence of a ZC45r CoV under culture conditions. In this regard, the phylogenetic discordance between NSP4 and NSP10, and RdRp is unlikely to have arisen due to sequence divergence under culture conditions, and appears more consistent with a novel sarbecovirus. In addition, the divergence of the RdRp of GX_ZC45r-CoV from bat-SL-CoVZC45 (0.037 nucleotide substitutions per site, Figure 8) seems too large to have occurred via culturing of a bat-SL-CoVZC45 isolate.

As several reads in the game animals datasets were found to map seamlessly from the 3′ end of NSP10 to the 5′ end of the RdRp of a novel SARS2r-CoV (Supplementary Figure S23), the simplest explanation is that NSP10 and RdRp form a contiguous section of a single genome, which as discussed above appears to have been in the form of cDNA plasmids when sequenced. The recovered NSP4 and NSP10 regions of GX_ZC45r-CoV, as well as a 297 nt section within the RdRp coding region all form a basal sister relationship to GX PCoVs. However, the RdRp coding region taken as a whole groups with the bat-SL-CoVZC45/HKU3-1/Longquan-140 clade on the SARS branch of a maximum likelihood phylogenetic tree (Figure 8). This is perplexing and could be explained if either the GX_ZC45r-CoV was a result of natural recombination, or if the genome was artificially manipulated to combine different parts of other genomes.

Although a 21 nt section matching the MCS of pUC57 plasmid was found in a read mapping to GX_ZC45r-CoV, it is unknown if any of the other synthetic vectors identified here are associated with GX_ZC45r-CoV laboratory research. While it is plausible that the vector containing SV40 promoter and Neomycin marker and the vector containing WPRE and puro sequences could be related to GX_ZC45r-CoV research, no virus sequences were found attached to these vectors and as such their research usage remains unknown.

Sequences with a 99.7% identity to HPIV2 strain t146a293_HPIV2 (MH892406.1) were identified in SRA datasets from multiple different animal species sequenced by He et al. [24,31]. That the same HPIV2 strain so closely matched a human strain, was found in multiple animal species, was associated with human genomic contamination and was found to comprise 28% of sample MJ-ZJ-MO-3 almost certainly indicates the virus was not related to a natural infection of pangolins as proposed by He et al., but instead, stems from laboratory contamination [24]. This finding is significant as He et al. propose the identification of HPIV2, previously considered to be only human hosted, in pangolins, civets and bamboo rats.

That GX_ZC45r-CoV exhibits strong phylogenetic grouping with the GX PCoVs in the NSP4, and NSP10 regions and grouping with an ancestral sequence in the partial RdRp region, raises questions as to the origin of the GX PCoV clade. If GX_ZC45r-CoV is a natural bat-hosted CoV, given the RdRp similarity to bat-SL-CoVZC45 and high identity of a 407 nt region of the RdRp to several Zhoushan bat CoVs sampled by Hu et al., the likely host species is *R.pusillus* located in Zhoushan city on Zhoushan Island, Zhejiang Province (Figure 1) [13]. If Malayan pangolins captured in Guangxi province [26] were infected with a bat-hosted virus originating from Zhoushan Island, the question then arises as to how did this occur? Furthermore, if the pangolins were smuggled from outside China the most likely country of origin is Vietnam [75], further distancing the pangolins from Zhejiang province.

It is of further concern that out of nine published GX PCoVs, only one unfiltered/non highly enriched pangolin tissue SRA dataset has been provided to support assembly of a GX PCoV, GX_P3B, a partial genome with 86% coverage of GX_P2V [24,25]. The dataset is of low quality with read lengths highly skewed to very short lengths, and is contaminated with SARS-CoV-2 reads. One other unfiltered/non highly enriched SRA dataset supporting GX PCoV assembly, GX_P2V a Vero-E6 cell culture sample, is also contaminated with SARS-CoV-2 reads [24].

Cross-contamination of samples [76,77] and contamination risks for metagenomic NGS projects are well documented [78,79]. Notwithstanding this, it is apparent that contamination at multiple potential stages prior to sequencing [78] and during sequencing via sample index hopping [80] is still a widespread problem [24,25,81]. We suggest that the occurrence of contamination is higher than commonly realized and warrants more attention. One problem with NGS contamination is the potential for misattribution of viral hosts, as we document for HPIV2 by He et al. We outline an example methodology which could be incorporated as part of quality control during NGS sequencing, and where anomalies are detected, re-sequencing rather than data filtering is recommended. Easily implementable checks include: alignment of reads using a kmer approach such as with fastv to a microbial database, alignment of de novo assembled contigs to NCBI databases, identification of synthetic vector sequences using code provided here, and use of a systematic mitochondrial mapping pipeline. These steps provide methods for detecting contamination by microbial and eukaryotic sequences, which may allow improvements to be made in extraction and sequencing protocols.

Finally, we request that He et al. publish the full GX_ZC45r-CoV genome, and document the sampling and sequencing history of this CoV. We further ask He et al. to document the source of and processes that led to the contamination of BioProjects PRJNA793740 and PRJNA795267 with human genetic material, HPIV2, GX_ZC45r-CoV and synthetic vectors. How pangolins smuggled into China and captured in Guangxi province came to be infected with a CoV highly related to a bat CoV from Zhoushan Island is perplexing. High quality, unfiltered datasets, free of contamination, of infected animals should be provided to the international community to ascertain the veracity of true pangolin infection with GX PCoVs.

## 5. Conclusions

Here, we analyzed a novel SARS2r-CoV first identified by Jones et al. [24], GX_ZC45r-CoV, and found that sections of the partial genome share the same ancestor as the GX PCoV clade. However, the RdRp phylogenetically groups with Zhoushan bat CoV bat-SL-CoVZC45, and a partial section of the RdRp groups with multiple Zhoushan bat SARS2r-CoVs. As such, it is possible the novel CoV is a *R.pusillus* hosted virus from Zhoushan Island, Zhejiang province, China. We identified the novel SARS2r-CoV in 7 additional game animal RNASeq datasets not previously identified, for a total of 16 datasets from 5 different species, all of which contain significant *H.sapiens* genetic material, and numerous viruses not associated with the host animals sampled. We further identified the dominant human haplogroup of the contaminating *H.sapiens* genetic material as F1c1a1, which is of East Asian provenance. Reads mapping to the novel CoV genome, in all animal datasets were solely located in the NSP4, NSP10 and RdRp coding regions. The common mapping locations, marked truncation of read coverage at two locations in the genome, and the presence of a multiple cloning site sequence from a pUC57 plasmid at the 5′ end of a read mapping to the NSP10 region are consistent with the presence of a laboratory virus sequenced from plasmids, rather than from a natural infection of the animals sampled. The novel virus has important evolutionary implications for the GX PCoV clade and we request that He et al. [31] publish the complete genome and sampling details to help elucidate the origin of the GX PCoV clade of viruses.

**Supplementary Materials:** The following supporting information can be downloaded at: https://www.mdpi.com/article/10.3390/applmicrobiol2040068/s1. Figure S1. SARS-CoV-2 spike protein amino acid similarity to SARS2r-CoVs. Figure S2. PCoV GX_P4L spike amino acid similarity plot to SARS2r-CoVs. Figure S3. Reads per sample aligning to GX_ZC45r-CoV gap-filled genome. Figure S4. Alignment of GX_ZC45r-CoV to bat-SL-CoVZC45 (MG772933.1) showing anomalous read coverage across position 14056nt potentially indicating artificial splicing. Figure S5. Alignment of GX_ZC45r-CoV to bat-SL-CoVZC45 (MG772933.1) showing anomalous read coverage across position 14758nt potentially indicating artificial splicing. Figure S6. Mitochondrial genome coverage >=10% for the 16 game animal datasets containing SARS2r-CoV reads. Figure S7. Blastn analysis of the recovered NSP10 region of GX_ZC45r-CoV showing the 13 highest identity genome matches. Figure S8. Simplot analysis of selected SARSr-CoV genomes using PCoV MP789 as a query. Figure S9. Simplot analysis of selected SARSr-CoV genomes using GX PCoV group as a query. Figure S10. Simplot analysis of four sections of PCoV MP20 which overlap with sections of GX_ZC45r-CoV after multi sequence alignment. Figure S11. Partial RdRp section (297nt) maximum likelihood phylogenetic tree using a GTR+I+G (GTR+FO+I+G4m) model with 1000 bootstrap replicates using raxmlGUI 2.0. Figure S12. Partial RdRp section (407nt) maximum likelihood phylogenetic tree using a T92+I model with 100 bootstrap replicates using MEGA11. Figure S13. Potential recombination region 1 detected using SiScan model in RDP5. Note 5' end of recombination region not detected. Figure S14. Potential recombination region 2 detected using SiScan model in RDP5. Figure S15. Partial vector sequence identified in MJ-ZJ-F-1 with similar layout to mammalian expression vector pSV2neo. Figure S16. Partial synthetic vector de novo assembled from the MC-GX-A-1 dataset. Figure S17. Synthetic plasmid sequence de novo assembled from the MJ-ZJ-F-3 dataset. Figure 18. Counts per read for reads mapping to selected viruses per SRA dataset for 46 SRA datasets from PRJNA793740 and PRJNA795267. Figure S19. Box plot distribution of counts per read mapping to selected viruses per SRA dataset for 46 SRA datasets from PRJNA793740 and PRJNA795267. Figure S20. Percent coverage of selected viruses by reads for 46 SRA datasets from PRJNA793740 and PRJNA795267. For all but GZ_ZC45r-CoV and SV40 a 10% cutoff was applied whereby each virus is present in at least one of the 46 datasets with 10% or greater genome coverage. Figure S21. Spearman correlation coefficient matrix for virus read counts for 46 SRA datasets from PRJNA793740 and PRJNA795267 (see Supplementary Figure S18 for sample names). Figure S22. Percent coverage of selected viruses by contigs for 64 SRA datasets from PRJNA793740 and PRJNA795267. SRA datasets were de novo assembled then aligned to a NCBI viral reference set. Figure S23. After alignment of the 16 datasets containing GX_ZC45r-CoV sequences to the bat-SL-CoVZC45 reference, 21 reads completely span the non-coding region between NSP10 and RdRp coding regions (indicated by vertical dashed lines). Figure S24. Maximum likelihood phylogenetic trees generated using PhyMLwith model selected using smart model section for: (a) partial NSP4 region using GTR+G+I model; (b) NSP10 region using TN93+G model; and (c) RdRp region using GTR+G model. Support values of 70% or greater are shown. Figure S25. Maximum likelihood phylogenetic trees generated using PhyML with model selected using smart model section for: (a) 297nt partial RdRp region using HKY85+G model; (b) 407nt partial RdRp region using T93+G model. Support values of 70% or greater are shown. Table S1. Haplogroup analysis of human mitochondrial reads identified in datasets with high GX_ZC45r-CoV presence. Table S2. Potential recombination breakpoints detected using RDP5 detected by more than three methods.

**Author Contributions:** Conceptualization, A.J., D.Z., Y.D. and S.C.Q.; Methodology, A.J., D.Z. and S.E.M.; Software, S.E.M. and A.J.; Formal Analysis, A.J., D.Z. and S.E.M.; Investigation, A.J., S.E.M., D.Z., Y.D. and S.C.Q.; Data Curation, A.J., D.Z. and S.E.M.; Writing—Original Draft Preparation, A.J.; Writing—Review and Editing, A.J., S.E.M., D.Z., Y.D. and S.C.Q.; Visualization, A.J. All authors have read and agreed to the published version of the manuscript.

**Funding:** This research received no external funding.

**Institutional Review Board Statement:** Not applicable.

**Informed Consent Statement:** Not applicable.

**Data Availability Statement:** Supplementary Info 1 can be found at: https://github.com/semassey/Scanning-NGS-datasets-for-mitochondrial-and-coronavirus-contaminants/blob/main/Mito-mappings-16-datasets-GX_ZC45r.zip (accessed on 9 October 2022); Supplementary data containing the files listed below can be accessed at: https://zenodo.org/record/7260031 (accessed on 9 October 2022) (doi:10.5281/zenodo.7260031): Supp_Info_2.xlsx, Supp_Info_3.xlsx; Novel GX_ZC45r-CoV genome: GX_ZC45r-CoV.fa; Gap filled GX_ZC45r-CoV genome: GX_ZC45-CoV_ZC45_gap_filled_no_polyA.fa; Read alignments to gap filled GX_ZC45r-CoV genome using minimap2: GX_ZC45-CoV_ZC45_gap_filled_no_polyA_minimap2_16_SRA.sam.

**Acknowledgments:** We thank Jonathan Latham for feedback which helped improve the manuscript.

**Conflicts of Interest:** The authors declare no conflict of interest.

**Source Code:** *Systematic mitochondrial mapping procedure.* The pipeline used for mapping NGS reads to all mitochondrial genomes in the NCBI database, and calculating their relative coverage (thus detecting contaminating eukaryotic species in an NGS dataset), can be found at: https://github.com/semassey/Scanning-NGS-datasets-for-mitochondrial-and-coronavirus-contaminants (accessed on 9 October 2022). *Primer search code* https://github.com/bioscienceresearch/Forensic_analysis_of_novel_SARS2r-CoV (accessed on 9 October 2022).

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
