# Peer review of "Forensic Analysis of Novel SARS2r-CoV Identified in Game Animal Datasets in China Shows Evolutionary Relationship to Pangolin GX CoV Clade and Apparent Genetic Experimentation"

_2673-8007, doi:10.3390/applmicrobiol2040068_

Round 1
Reviewer 1 Report
The article presented by Jones A et al., examines datasets of different game animals and concludes that the presented regions have more similarity to sequences with pangolin origin and are likely contaminants or genetic experimentation. The manuscript is well written and the results are consistent.
The conclusions of the study are quite interesting and they are well supported by the performed analysis. I have only a few technical remarks:
Please, describe in more details how the phylogenetic analysis was performed and why the authors used MEGA-X instead of more robust phylogenetic pipeline, instead of constructing approximate ML-trees.
In the discussion section, talk a little bit more over contamination in next-generation sequencing and how it can influence the interpretation of the data, as this is the case in this article. How contamination can be avoided and what is the best bioinformatic pipeline in these situations?
Author Response
Please, describe in more details how the phylogenetic analysis was performed and why the authors used MEGA-X instead of more robust phylogenetic pipeline, instead of constructing approximate ML-trees.
> We repeated the analyses using PhyML, and find that MEGA-X accurately recovers ML trees (the trees are highly similar to each other). Consequently, the use of MEGA-X is justified, We have described the PhyML analyses in the revised MS as follows:
Page 13: "A maximum likelihood tree implemented using PhyML using a GTR+G+I model shows the same basal sister relationship of GX_ZC45r-CoV to GX PCoVs (Supp. Fig. 24a)."
Page 14: "Repeat analysis using PhyML with a TN93+G model shows the same relationships (Supp. Fig. 24b)."
Page 15: "A maximum likelihood tree using PhyML with a GTR+G model also shows GXZ_ZC45r-CoV on the same branch as bat-SL-CoVZC45 (Supp. Fig. 24c)"
In the discussion section, talk a little bit more over contamination in next-generation sequencing and how it can influence the interpretation of the data, as this is the case in this article. How contamination can be avoided and what is the best bioinformatic pipeline in these situations?
> A section was added to the Discussion, as follows:
"Cross-contamination of samples [85] [86] and contamination risks for metagenomic NGS projects are well documented [87] [88]. Notwithstanding this, it is apparent that contamination at multiple potential stages prior to sequencing [87] and during sequencing via sample index hopping [89] is still a widespread problem [25] [26] [90] [91]. We suggest the occurrence of contamination is higher than commonly realized and warrants more attention. One problem with NGS contamination is the potential for misattribution of viral hosts, as we document for HPIV2 by He et al. [37] should be paid to it. We outline an example methodology which could be incorporated as part of quality control during NGS sequencing, and where anomalies are detected, re-sequencing rather than data filtering is recommended. Easily implementable checks include: Alignment of reads using a kmer approach such as with fastv to a microbial database, alignment of de novo assembled contigs to NCBI databases, identification of synthetic vector sequences using code provided here, and use of our systematic mitochondrial mapping pipeline. These steps provide methods for detecting contamination by microbial and eukaryotic sequences, which may allow improvements to be made in extraction and sequencing protocols."

Reviewer 2 Report
The article entitled "Forensic analysis of novel SARS2r-CoV identified in-game animal datasets in China shows the evolutionary relationship to Pangolin GX CoV clade and apparent genetic experimentation." is a very interesting piece of research. The results and discussion are presented in a very well mannered. I read the article with great interest, and this research is of significant importance. However, some suggestions should be considered before the article's publication.
Line no. 30: A zoonotic jump from animals has been proposed as a potential origin for SARS-CoV-2. This statement needs a recent reference such as:
[https://doi.org/10.1016/j.ijsu.2021.106208]
Further, the emergence of other lineages of the SARS-CoV-2 can be mentioned, and their link with the zoonotic spillover can be briefly discussed from the article: [https://doi.org/10.1016/j.biopha.2022.113522]. Such as:
Coronaviruses (CoVs) are known for their capacity to traverse species boundaries, and new research suggests that SARS-CoV-2 may be found in a variety of species, including home pets, commercial animals along with wildlife which strongly suggests the animal spillover of the SARS-CoV-2 in a potential host and accumulation of a large number of mutations in the animal host and jumps back into the human population [https://doi.org/10.1016/j.biopha.2022.113522].
Line no. 31 to 32: Provide the year for clarity
Line no. 38: lineage B, for a reader, it is difficult to associate. Hence, I suggest explaining about lineage A and B in brief.
Line no. 41 to 43: I suggest fragmenting the information to make it more clear.
I suggest reducing the complexity of the information, especially in the Introduction section, as it will increase the reachability of your article.
Line no. 84: Coronaviruses (CoVs)
Line no. 597: We further identify, change to identified. Likewise, I suggest checking the whole article for minor grammatical and English errors that can be easily overlooked.
I congratulate the authors for such important research.
Best Wishes.
Author Response
Line no. 30: A zoonotic jump from animals has been proposed as a potential origin for SARS-CoV- 2. This statement needs a recent reference such as:
[https://doi.org/10.1016/j.ijsu.2021.106208]
> An appropriate reference has now been inserted
Further, the emergence of other lineages of the SARS-CoV-2 can be mentioned, and their link with the zoonotic spillover can be briefly discussed from the article: [https://doi.org/10.1016/j.biopha.2022.113522]. Such as:
Coronaviruses (CoVs) are known for their capacity to traverse species boundaries, and new research suggests that SARS-CoV-2 may be found in a variety of species, including home pets, commercial animals along with wildlife which strongly suggests the animal spillover of the SARS- CoV-2 in a potential host and accumulation of a large number of mutations in the animal host and jumps back into the human population [https://doi.org/10.1016/j.biopha.2022.113522].
> Thankyou for this suggestion, but we wanted to streamline the Intro, as per your recommendation below and so decided not to include this discussion
Line no. 31 to 32: Provide the year for clarity
> This has been done
Line no. 38: lineage B, for a reader, it is difficult to associate. Hence, I suggest explaining about lineage A and B in brief.
> This has been done as follows:
" In addition, the presence of only lineage B associated with human infections at the market [6] makes a market origin less likely, as lineage B is likely a derived lineage, while lineage A is likely ancestral [8].( lineage A and lineage B are the first two major SARS-CoV-2 lineages to emerge in 2019, and are only separated by two single nucleotide variants (SNVs)"
Line no. 41 to 43: I suggest fragmenting the information to make it more clear.
> We have clarified the text in this section
I suggest reducing the complexity of the information, especially in the Introduction section, as it will increase the reachability of your article.
> We have reduced the size of the Introduction, and have streamlined the text throughout
Line no. 84: Coronaviruses (CoVs)
> This has been done
Line no. 597: We further identify, change to identified. Likewise, I suggest checking the whole article for minor grammatical and English errors that can be easily overlooked.
> This change has been made, and we have thoroughly checked for and corrected grammatical and English errors throughout
